# Zinc’s Association with the CmPn/CmP Signaling Network in Breast Cancer Tumorigenesis

**DOI:** 10.3390/biom12111672

**Published:** 2022-11-11

**Authors:** Mellisa Renteria, Ofek Belkin, Justin Aickareth, David Jang, Majd Hawwar, Jun Zhang

**Affiliations:** Department of Molecular and Translational Medicine (MTM), Texas Tech University Health Science Center El Paso, El Paso, TX 79905, USA

**Keywords:** Zinc, CmPn signaling network, CmP signaling network, CCM signaling complex (CSC), progesterone (PRG), classic nuclear progesterone receptors (nPRs), non-classic membrane progesterone receptors (mPRs/PAQRs)

## Abstract

It is well-known that serum and cellular concentrations of zinc are altered in breast cancer patients. Specifically, there are notable zinc hyper-aggregates in breast tumor cells when compared to normal mammary epithelial cells. However, the mechanisms responsible for zinc accumulation and the consequences of zinc dysregulation are poorly understood. In this review, we detailed cellular zinc regulation/dysregulation under the influence of varying levels of sex steroids and breast cancer tumorigenesis to try to better understand the intricate relationship between these factors based on our current understanding of the CmPn/CmP signaling network. We also made some efforts to propose a relationship between zinc signaling and the CmPn/CmP signaling network.

## 1. Introduction

Despite recent scientific advancements in cancer research, breast cancer remains one of the leading causes of cancer-related deaths globally and is expected to increase in parallel with the upward projection of the elderly population [1,2,3]. It is estimated that the majority of breast cancer subtypes are responsive to sex steroid hormones (particularly estrogen and progesterone) due to their expression of classic estrogen receptors (ERs), classic progesterone receptors (nPRs), or both, and are therefore termed hormone-responsive cancers [4,5]. Additional steroid hormone receptors, such as non-classic ERs/PRs, have been discovered in recent advances in the field [6,7,8,9]. In comparison, triple-negative breast cancer (TNBC) is a distinct form of breast cancer in which cells lack expression of ERs, nPRs, and epidermal growth factor receptor 2 (HER2) [10]. As a result, TNBC is considered a non-responsive breast cancer due to lack of sex steroid hormone-receptor expression and this property of TNBCs is largely responsible for the disease’s aggressive nature, high metastatic potential and worse prognosis, as well as the limited therapeutic options [11,12]. Various properties that contribute to efficacious metastasis include cell motility, invasion, plasticity, and microenvironmental modulation [13].

Zinc, one of the essential micronutrients, is involved in a variety of cellular processes such as structural or catalytic components in an assortment of proteins and has been noted to be involved in these metastatic processes as well [13,14]. Zinc is required for the action of more than 300 metalloenzymes, 3000 transcription factors, and thousands of other cellular proteins, which play crucial roles in physiological processes, including antioxidant, anti-inflammatory and immune responses, as well as apoptosis [14,15]. Furthermore, zinc plays an important role in the polymeric organization of macromolecules such as DNA and RNA, protein synthesis, and cell division [16]. In addition, an estimated 10% of human proteins are known or predicted to bind zinc ions [17]. Aberrant zinc levels may lead to various human conditions, including cancers [18]. However, zinc supplementation could ameliorate the inflammation in cancer [18]. Unfortunately, research on the regulatory effect and influence of zinc deficiency during tumorigenesis remains elusive. As zinc supplementation has gradually been applied in nutritional support therapy for cancer patients, further exploration of the regulation of enzyme activity by zinc deficiency in cancer may provide comprehensive guidance for the application of zinc therapy [19]. Previous studies demonstrated that zinc concentration is especially important in the microenvironment modulation of tumor cells’ acquisition of tumorigenic potential [13]. Additionally, there is evidence that zinc dyshomeostasis caused by a dysfunctional zinc transporter can contribute to the initiation of various cancers such as prostate, breast, and pancreatic cancers [20]. Serum and intracellular zinc concentrations are altered in breast cancer patients, and it remains unclear if this redistribution is a direct consequence of zinc nutrient alteration or a tissue-specific response [21]. Zinc signaling directly controls mammary gland development and aberrant signaling has been observed in malignant cells during breast cancer progression, accumulation, or the redistribution of zinc in the mammary gland [22].

## 2. Cellular Zinc Level Is Influenced by Dietary Supplementation

### 2.1. Cellular Zinc Concentrations Depend on Dietary Consumption 

Minerals and other trace elements are essential micronutrients for normal physiological functioning and well-being [3]. The daily intake and homeostasis of these micronutrients are largely dependent on dietary habits [3]. Although zinc is the most abundant intracellular micronutrient, the human body is incapable of storing sufficient amounts of zinc; therefore, an inadequate diet can rapidly lead to zinc deficiencies [15,23]. It is estimated that nearly 50% of United States adults over the age of 50 are consuming suboptimal amounts of zinc and nearly two billion people worldwide may be zinc-deficient [15]. In rodents, nutritional zinc deficiencies have shown to predispose subjects to a reversible carcinogenesis, in which the subsequent replenishment of dietary zinc led to a reduction in cellular multiplicity and subsequent progression of the malignancy, likely by altering cellular proliferation and gene expression [24,25]. Epidemiological studies have shown that an inverse relationship exists between dietary zinc consumption and the development of breast cancer [23,26]. Specifically, evidence shows that biopsies of breast tumor cells contained significantly higher intracellular zinc concentrations and increased zinc transmembrane protein expression compared with normal tissue cells [26]. Therefore, the aberrant expression and homeostasis of zinc in breast tumors correlates with malignancy and could contribute to the severity of this cancer subtype [26].

### 2.2. Zinc Cellular Specific Actions

Zinc has shown to modulate immune system functioning, as well as the regulation of various metabolic, genetic, and cell-signaling pathways [14,15]. Studies have shown that zinc plays a protective role in tumor initiation and development by reducing oxidative stress and protecting DNA from reactive oxygen species (ROS) and the subsequent development of oncogenic mutations [27]. Specifically, zinc’s function as an antioxidant provides genomic stability by decreasing oxidative DNA damage [26]. However, other studies suggest that cytotoxic levels of zinc are also known to cause DNA damage, oxidative stress, and the formation of ROS [20]. Several findings suggest that zinc’s function is largely concentration- and cell-specific and this property of zinc may result in both pro-apoptotic and anti-apoptotic properties of the micronutrient [14]. While zinc has been shown to play a role in numerous malignancies, as a result of the complex nature of zinc homeostasis, a delineated relationship between zinc and tumorigenesis has yet to be established [14]. 

## 3. Zinc Plays a Significant Role in Tumorigenesis

### 3.1. Function of Zinc Contributes to the Progression of Cell Tumorigenesis

The biological effects of zinc can be exerted through the intra- and extracellular zinc regulatory functions and its interactions with proteins [28]. Zinc can act as either extracellular stimuli or intracellular messengers. Therefore, a precise working model of zinc regulatory mechanisms is needed to obtain a better understanding of homeostatic control for transients, subcellular distribution and trafficking, organellar homeostasis, and vesicular storage and exocytosis of zinc ions [29]. The vast majority (95%) of zinc is located intracellularly so that the extracellular concentration available is low. Intracellular zinc homeostatic molecules include cytosolic zinc-binding proteins, transporters localized to cytoplasmic and organellar membranes, and sensors of cytoplasmic free zinc ions [30]. Circulatory zinc is mainly bound to albumin, transferrin, and α2-macroglobulin but remains accessible to zinc transporters to control the cellular zinc balance [31]. Intracellular levels of zinc are largely coordinated by zinc transport channels, which are capable of both zinc influx and efflux [15]. There is evidence suggesting that zinc accumulates in elevated levels in breast cancer cells and other malignant cell lines in comparison to normal mammary epithelial cells [32]. Specifically, breast cancer cells showed a 72% increase in intracellular zinc concentrations in comparison to other, non-malignant breast cells, while serum levels of zinc were decreased from baseline [33]. One meta-analysis of 36 studies containing more than 5700 patients found significantly decreased serum zinc levels in breast cancer patients in comparison to controls and patients with benign breast diseases [34]. The findings of increased intracellular zinc concentrations were also observed at the histological level, suggesting that cellular zinc concentrations may be clinically useful in determining malignancy grading, as well as serving as predictive biomarkers for breast cancers [35]. However, the exact mechanisms that are responsible for the accumulation and dysregulation of zinc in breast tumors are not well understood and it remains unclear as to whether intracellular zinc accumulation causes the disease or is a consequence of this disease [15,32].

### 3.2. Zinc Transport Protein in Breast Cancer Cells

While the exact mechanisms of zinc’s role in tumorigenesis is not well-understood, there are many speculations. Zinc acts as a signaling molecule, and both its intracellular and extracellular concentrations must be tightly regulated for proper physiological functioning [36]. Therefore, there is a complex regulatory system for the precise homeostatic control of cellular zinc transport, distribution, trafficking, organelle homeostasis, vesicular storage, and exocytosis of zinc ions [29,37]. There are several regulators of free intracellular zinc, such as zinc transporters, inhibitory factors, and sensors. Among the zinc transporters, Zrt-/Irt-like proteins (*ZIPs*) are most frequently studied [38,39,40]. Cellular zinc levels are strictly controlled by two families of transport proteins: ZIP channels (SLC39A) and ZnT transporters (SLC30A). ZIP channels increase cytosolic zinc levels by importing zinc into cells or releasing zinc from endoplasmic reticulum (ER) [41,42,43]. One subfamily of ZIP, estrogen-regulated LIV-1 (SLC39A6) has been implicated in breast cancer [42,44]. For example, the gene expression levels of LIV-1, a membranous zinc transporter, have been shown to increase four-fold under exposure to estrogen treatment [23,24]. Similarly, increased expression levels of LIV-1 were also observed under progesterone (PRG) treatment [23]. LIV-1 is one of the few zinc transporters identified to contain a metalloproteinase motif, which is responsible for breaking down the basement membrane and allowing for the metastasis of breast cancer cells [25]. Likewise, studies have shown that the expression levels of another zinc transporter, ZIP10, were significantly increased in metastatic breast cancer cell lines (such as MDA-MB-231 and MDA-MB-435S) when compared to less invasive cell lines (such as T47D, MCF7, ZR75-1, and ZR75-30). In addition, attenuating ZIP10 or intracellular zinc chelation in MDA-MB-231 cell lines lead to the inhibition of malignant cell migration [18,26]. Furthermore, ZIP7 was found to be able to release zinc from the ER, which leads to zinc-mediated tyrosine kinase signaling to activate cell migration and growth. This result suggests that ZIP7 might be a novel therapeutic target for breast cancer [45]. Contradicting results for the roles of LIV1 in breast cancer tumorigenesis have been reported: some studies demonstrate that ER-positive(+) breast cancer cells, which, according to the aforementioned mechanism above, have increased LIV-1 and intracellular zinc levels, are associated with better outcomes [46], as they are responsive to anti-estrogenic therapies such as Tamoxifen and aromatase inhibitors [27]. Similarly, ZIP6 deficiency disturbs intracellular Zn(2+) homeostasis, leading to increased cell survival [44]. However, breast samples from patients showed significant increases in both ZIP7 and ZIP6 in tumors, and the Kaplan–Meier curve revealed that high ZIP7 levels are correlated with decreased overall survival of patients [47]. These contradicting results can be explained by the tumor-specific hormonal response for different members of ZIP. Another cellular zinc transport receptor, ZnT2, has been shown to function in zinc sequestration, protecting cells from the cytotoxic effects of excess intracellular zinc, ROS formation, and subsequent apoptosis [28]. This study demonstrated that increased expression levels of ZnT2 transporters in malignant breast cancer cells protects these cells from apoptosis and that, conversely, tumor cells with decreased expression levels of ZnT2 transporters were less viable [28]. As a result, attenuating intracellular zinc-sequestering mechanisms may be a viable strategy for treating malignant breast cancers [28]. One study displayed a correlation between zinc concentration and histological and molecular grading and subtypes, showing elevated zinc levels in TNBC [29]. This study also displayed that increased intracellular levels of zinc were correlated with increased aggressiveness of breast cancers, with the highest zinc concentrations being present in HER2-positive breast cancers and TNBCs [29]. Additional zinc cellular regulatory factors are zinc inhibitory factor (ZIF) and zinc-sensing G-protein coupled receptor (ZnR/GPR39). ZIF reduces free intracellular zinc by inhibiting zinc transport in the oocyte before ovulation [48]. As a G-protein coupled receptor, ZnR/GPR39 triggers intracellular Ca^2+^ release and subsequently activates downstream MAPK or PI3K/AKT pathways controlling cell proliferation [49]. ZnR/GPR39 activity has been found to be enhanced in breast cancer [50,51,52].

## 4. Zinc and the CmPn/CmP Signaling Network

### 4.1. Zinc Is a Critical Nutrient in Mammalian Female Reproductive System

Although zinc has been recognized as a critical nutrient in mammalian female reproduction, its specific role in the female reproductive system was only recently recognized [53,54,55]. Accumulated data indicates that zinc depletion causes multiple defects in the female reproductive system in mice [56]. Zinc deficiency in females can lead to series of reproductive problems, including impaired synthesis and secretion of follicle stimulating hormone (FSH) and luteinizing hormone (LH), abnormal ovarian development, disruption of the estrous cycle, increased risk of abortion, prolonged gestational periods, teratogenicity, stillbirths, difficulty in parturition, pre-eclampsia, toxemia, and low fetal birth weights [57]. In addition, it is well known that zinc deficiency during pregnancy in experimental animals results in fetal anomalies [58,59,60]. Furthermore, additional data indicated that maternal supplementation of zinc drastically improved female pregnancies in sheep [61]. However, this conclusion has not yet been fully validated in humans [62].

### 4.2. Zinc Supplementation for Cancer Prevention

Due to its crucial functions and effects on multiple processes in the tumorigenesis and progression of cancers, the potential impact of zinc supplementation on cancer prevention and intervention as an anticancer and antitoxicity agent cannot be ignored [63,64,65]. To date, some notable progress has been made in this area [66]. As an essential micronutrient in foods and mineral supplements, zinc has been examined in many clinical trials for the possible prevention of gynecological cancers [64], like breast [26] and ovarian cancers [65], gastrointestinal and hepatic cancers [67,68,69,70], and lung [71] and oral cancers [72], in addition to cardiovascular disease (CVD) [73,74]. However, if breast cancers sequester and use more zinc than normal cells, therapeutic use of an excessive amount of zinc needs to be examined in animal models before any attempt at clinical trials for cancer treatment.

### 4.3. Zinc Is an Essential Nutrient in PRG Biogenesis and PRG-Mediated Signaling

Zinc has been shown to function in the production of FSH and LH [75], which play a role in ovulation and the subsequent production of PRG. As a result, it has been determined that dietary zinc deficiencies have a negative impact on the developmental potential of oocytes [76,77], which could affect *corpus luteum* (CL) development and PRG production [78]. Likewise, serum zinc levels have been found to be significantly lower in PRG-associated reproductive disorders in sheep [79], suggesting that there is a positive correlation between zinc levels and PRG production. However, relevant reports on the relationship between zinc and PRG are not without controversy. One report determined that two weeks’ supplementation with low-dose zinc had no significant effect on serum estradiol and PRG concentration in postmenopausal women [80]. Another report suggested that zinc may play an inhibitory role in the onset and maintenance of PRG production in a mouse study. However, the increase in PRG under zinc-depleted conditions could be due to the removal of inhibitory pathways of zinc-mediated signaling [78], as signaling through the zinc-binding the SMAD (mothers against decapentaplegic homolog) transcriptional pathway is known to inhibit PRG production [81,82,83]. Therefore, the accumulated data indicate that the potential impact of zinc on the female reproductive system should not be ignored [75].

As mentioned prior, zinc is a key modulator of mammary gland development and maintenance [84]. Similarly, PRG and nPRs are responsible for the growth in alveolar epithelial structures, milk production, and secretion [85]. Clinical data indicate that PRG is a risk factor for breast cancer and that alterations in PRG-nPR signaling pathways contribute to early-stage human breast cancer progression [86]. Furthermore, it has been demonstrated that zinc can change the binding properties of PRG to its nPRs in the human endometrial cytosol [87]. Membrane progesterone receptors/progestin and adipoQ receptors (mPRs; mPRα-ε is identical to PAQR5-9) share structural similarities with other progestin and adipoQ receptors (PAQRs) that have a binding pocket for free fatty acid [88]. Protein structural data indicated that PAQR1 has an arginine residue (positive charge) in the binding pocket that is occupied with an oleic acid. PAQR1 was only able to bind PRG under the concentration of 100 µM zinc. Similarly, since mPRα (PAQR7) has the same binding pocket, the addition of high concentrations of zinc would form a salt with the fatty acid in the binding pocket, allowing for PRG to bind mPRα [89]. Therefore, sufficient data indicate that zinc can modulate the binding of PRG to either nPRs and/or mPRs [9,87], three key components of the CmPn (CSC-mPRs-PRG-nPRs) signaling network.

### 4.4. Zinc Associated with the CSC

Since the CSC (CCM signaling complex) [90,91,92,93] is the first component in newly defined CmPn (CSC-mPRs-PRG-nPRs)/CmP (CSC-mPRs-PRG) signaling networks in nPR (+/−) breast cancer cells [94,95,96], its relationship with zinc will be explored before other key components of the CmPn/CmP signaling networks. It has been reported that acute and subacute hemorrhage were found in head and neck cancers [97,98]. Furthermore, adenocarcinomas have shown to exhibit a depletion of zinc due to the down-regulated gene expression of hZIP1, a zinc transporter [99]. This indicates that local zinc deficiency in cancer lesions is a critical early event in tumorigenesis and can indirectly lead to unexpected brain hemorrhage. Therefore, a similar dietary supplementation strategy has been proposed for the prevention of CVDs, similarly to the proposed supplementation for cancer prevention [74]. However, contradicting in vitro and in vivo data demonstrated that both zinc deficiency and over-supplementation were associated with disintegration of the blood–brain barrier (BBB) leading to hemorrhagic stroke [100,101,102,103], implicating zinc as an independent risk factor for hemorrhagic stroke [103]. This once again emphasizes the importance of the tight regulation of intra- or extracellular zinc concentrations in proper homeostatic function [28,29]. The core components of the CSC are composed of three CCM proteins (CCM1, 2, 3) [93]. Deficiencies in any one of these will compromise microvascular integrity [92,93,104] and lead to hemorrhagic stroke in familial CCMs [90], suggesting a possible link between the CSC and zinc. Zinc localization and transporters were reported to be altered in Ccm1 zebrafish models and, in the same study, transcriptome analyses showed that solute carrier family 39 member 6, or LIV1 with 9 ZIPs, 4–8, 10, 12–14 (SLC39) zinc transporter genes [42] are misregulated in brain microvascular endothelial cells (BMECs) isolated from *Ccm1* null mice. These results indicate a perturbed transcription of SLC39 zinc transporter genes in the depletion of *CCM1* [105]. Subsequently, SLC39 zinc transporters have been linked to the CSC signaling cascades, and been defined as the direct downstream target of hyper-activated MEKK3–MEK5–ERK5 kinase cascade [105,106]. SLC39 is currently recognized and utilized as a key factor in the CSC signaling in various reports [107,108,109,110,111,112,113,114,115,116]. Ironically, all these reports relied on a single report that showed a “conserved CCM complex” (Ccm1/2 in the CSC) modulates ERK-5/MAPK-KLF-3 signaling factor to facilitate the expression of the SLC39 ortholog (or paralog, 24–30% identity in amino acid sequence between human and *C. elegans*/*Drosophila*), zipt-2.3, in *C. elegans*. A deficiency of Ccm1/2 will lead to abnormal zinc transporter (SLC39) expression and zinc storage in the intestinal granules [105]. Recently, comparative genomics in *vertebrate and invertebrate CCM* models have demonstrated that, despite the fact that CCM3 is evolutionarily conserved as an ancient gene (40–48% identity in amino acid sequence between human and its invertebrate counterparts, *C. elegans*/*D. melanogaster*), in contrast to a previous report [105], CCM1 and CCM2 orthologs have not been identified in *invertebrates* [117,118,119]. This suggests one of two possibilities: either the CCM1/2 gene-pair may be the latest evolutionary newcomers in the CSC or, more likely, this gene pair form the CSC first and later recruits CCM3 into this signaling complex that specializes in angiogenic endothelial cell (EC) maintenance in the closed circulatory system of *vertebrates* [117,119]. Unlike *vertebrates*, most organisms in invertebrates, such as arthropods (*D. melanogaster*), have an open circulation system; however, nematodes (*C. elegans*) have *no* circulatory system at all. This fact calls the existence of only an invertebrate ortholog of CCM1 deposited in the genebank into question (in fact, a short appearance of a drosophila ortholog of CCM1 was quickly removed from the genebank, and an in *vertebrate* CCM2 ortholog was never published), which could be an CCM1-irrelevant gene involving the signaling cascades affecting *C. elegans* lifespan [120,121]. Therefore, excessive caution should be applied when dealing with these kri-1/ SLC39 associated signaling pathways/cascades in the CSC mediated pathogenesis of CCMs [107,108,109,110,111,112,113,115].

Interestingly, more data demonstrate the functional association of zinc with the CSC signaling in an indirect fashion, such as zinc-containing superoxide dismutase (SODs) and related reactive oxygen species (ROS) [122,123,124,125], a downstream target of SODs, matrix metalloproteinases (MMPs) [126] and its counterpart, a tissue inhibitor of metalloproteinase-1 (TIMP-1) [127,128,129,130,131,132,133,134].

### 4.5. Involvement of Zinc within the CmPn/CmP Signaling Network

It is clinically well-known that a variety of breast cancer subtypes are hormone-responsive and that these hormones have potentially debilitating effects on the growth, metastatic properties, and subsequent prognosis of breast cancer subtypes. As a matter of fact, it is estimated that nearly 70% of breast cancers express ERs, nPRs, or both [4,5]. The exact intracellular physiological mechanisms by which these hormones function remains largely unstudied. In a previous study, we provided evidence suggesting that the CCM signaling complex (CSC) functions as a bridge for crosstalk among nPRs, mPRs, and their ligands, to form what we proposed to be the CSC-mPRs-PRG-nPR (CmPn)/CmP signaling network [91,92,94,95,96,135,136,137]. Studies have shown variable expression patterns of mediators of this pathway across various breast cancers, suggesting that this pathway is largely involved in breast tumorigenesis [94,95,96,136]. Some breast cancer subtypes expressing mPRs have partially contributed to the discovery of mPR-specific PRG actions of the CmP signaling network (CSC-mPRs-PRG). Furthermore, the findings of elevated zinc levels in TNBCs, zinc’s potential for modulating effects of PRG and nPRs, and the nucleocytoplasmic shuttling and localization properties of mPRs under the influence of zinc all call for further investigation into zinc and its function within the CmPn/CmP signaling network in the breast cancer tumorigenesis of TNBCs [138].

In this review, based on the presented evidence that zinc affects all key components of the CmPn/CmP signaling networks, we propose an intricate mechanism by which zinc functions within the CmPn/CmP signaling networks in modulating breast cancer tumorigenesis (Figure 1). We suggest that altered expression levels of CmPn/CmP mediators, as well as altered expression levels of ZIP channels (SLC39A) and ZnT transporters (SLC30A), and serum and intracellular zinc levels, may serve as diagnostic and prognostic biomarkers due to their shared involvement in the malignant transformation of breast cells [94,95,96,136]. In this review, we provided published evidence of a critical relationship between zinc and CmPn/CmP signaling networks, which may be responsible for the increased risks with breast cancers, necessitating further investigation into the physiologic dynamic between zinc and the CmPn/CmP signaling network (Table 1).

## 5. Conclusions

The relationship between serum and cellular concentrations of zinc and tumorigenesis is rather intricate, and the exact mechanism by which zinc promotes tumorigenesis is not well understood. Furthermore, many studies that have previously studied the relationship between zinc and oncogenesis failed to elaborate on whether their findings relate to serum or intracellular concentrations of zinc and the association with the potential initiation of tumorigenesis. This has led to many contradicting results. In this review, we attempted to summarize the critical associations between dynamic cellular levels of zinc during breast tumorigenesis in an attempt to elaborate on the relationship between the sex steroid hormones, estrogen and PRG, their corresponding receptors, and zinc. We summarize our review using our proposed model, which depicts the mechanisms by which the CmPn/CmP signaling complex and zinc modulate breast cell tumorigenesis.

## Figures and Tables

**Figure 1 biomolecules-12-01672-f001:**
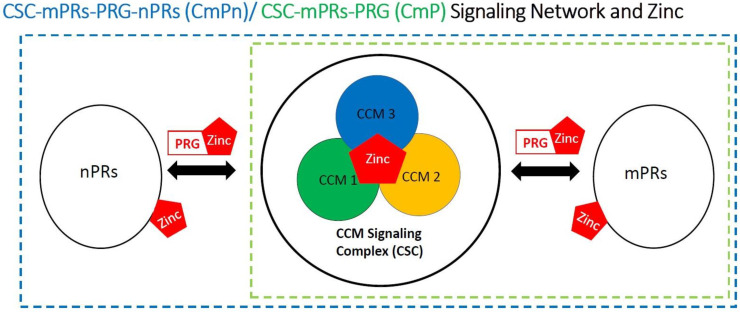
Signaling network bridge “crosstalk” among key players within CmPn signaling network. The diagram displays the relationship among PRG, nPR, mPR, CSC, and zinc (Zn). We show that the zinc interacts with all major players within the CmPn signaling network and involved in CmPn-modulating signaling during breast cancer tumorigenesis.

**Table 1 biomolecules-12-01672-t001:** Detailed information regarding the CmPn signaling network. A list of recent key publications describing CmPn and CmP signaling networks among nPR(+) and nPR(-) cancers and vascular ECs.

Main Point	Key Findings	Pubmed	References
The CSC is linked to tumorigenesis with mPRs in various cancers.	Differential expression of CCM and mPR correlated with various types and grades of major human cancers, especially breast and liver cancers.	**PMID:** 32186778	[136,138]
Establishing the CmPn signaling network in nPR(+) breast cancers	The CSC role in coupling classic, non-classic, or combined PRG signaling pathways via the effects of an intricate homeostatic concentration of progesterone to form the CmPn signaling network	**PMID:** 35971177	[96,138]
Establishing the CmP signaling network in nPR(-) breast cancers	Through establishingCmP signaling network in nPR(-) breast cancers, we discovered novel biomarker signature panels for Triple-Negative Breast Cancers (TNBCs) between African and Caucasian Women	**PMID:** 35431232; 35481969	[94,95,138]
Establishing the CmP signaling network in nPR(-) vasular ECs	Deficiency of any CCM genes, in combination with mPR-specific PRG actions, leads to perturbed CmP signaling network in nPR(-) ECs both in vitro and in vivo, result in compromising blood brain barrier integrity and increases the risk of hemorrhage	**PMID:** 36077089; 35098046	[135,137,138]
Exploring molecuular signaling within the CmPn/CmP signaling networks with multiomics	molecuular signaling within the CmPn/CmP signaling networks were investigated with multiomics, such as RNAseq and proteomics. Major molecular pathways were presented with pathway analysis and visualization	**PMID:** 36077089; 35098046	[91,94,95,96,135,136]

## Data Availability

Not applicable.

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
