# Peer review of "Zinc’s Association with the CmPn/CmP Signaling Network in Breast Cancer Tumorigenesis"

_biomolecules, 2022, doi:10.3390/biom12111672_

Round 1

Reviewer 1 Report

The paper by Renteria et al, summarises the latest findings about the association of the CmPn/CmP network to the altered expression of ZIP transporters and intracellular zinc which are involved in breast cancer tumorigenesis. This review adds to the current knowledge of the role played by zinc in the development of this disease, which has been reported in many investigations in the last decades. Overall, this is a good review covering different aspects of zinc biology, from its essential role as a dietary nutrient to its role in cellular homeostasis and diseases. In particular, the last paragraph about the link between zinc and the CmPn/CmP network is well discussed.

There is evidence that zinc can promote cellular migration and increase cell growth, which is related to zinc's ability to inhibit tyrosine phosphatases. This could have been highlighted when discussing the role zinc plays in tumorigenesis.  Furthermore, in paragraph 3, it is mentioned that LIV-1 overexpression and increased intracellular zinc are associated with better outcomes in breast cancer because cells become more responsive to anti-oestrogen therapy. There are, however, a few reports that have now shown that overexpression of members of the LIV-1 subfamily, including ZIP6, has been associated with worse cancer prognosis and endocrine resistance, a serious concern that cannot be overlooked in the treatment of cancer.

Nevertheless, I think that altogether the review is well written and therefore I recommend it for publication.

Author Response

Please see attached letter

Reviewer 2 Report

Excellent review of current studies and good with valid model. Future prospective studies needed

Author Response

Please see attached letter

Reviewer 3 Report

This review article focuses on reviewing the role of zinc in breast cancer, and ties in the author’s previous work showing the connection between PRG signaling and the CCM signaling complex. As several recent reviews on zinc homeostasis and breast cancer have been published, the novelty of this review is the author’s attempt to bring in the so called “CmPn/CmP signaling network”. However, this attempt is heavy handed and lacks a clear rationale.

Specifically, the section “Zinc associated with the CSC” starting on line 230 is poorly organized and doesn’t lay out a compelling rationale. There are statements presented without evidence (“This indicates that local zinc deficiency in cancer lesions is a critical early event in tumorigenesis and can indirectly lead to unexpected brain hemorrhage.” Line 233); experimental design of others is misrepresented (“Therefore, efforts to directly link zinc to CCM conditions have been made.” Line 244); and several tangents are taken that seriously confuse the overall argument that the authors appear to be trying to make (i.e. “comparative genomics lines 254-262, SOD lines 263-273, and MMPs/ROS lines 273-310). A major reworking of this section (4) to clarify 1) the speculative hypothesis being proposed and 2) the individual lines of evidence that support this hypothesis is needed. As the majority of the review focuses on zinc and breast cancer, the authors need to clearly connect their hypothesis to this disease, and be transparent about the limitations of interpreting data from other cell/cancer types. Finally, the last paragraph of section 4 contains several conclusions not supported by the review, including that “direct evidence for a critical relationship by which estrogen and PRG are directly responsible for the increased intracellular zinc levels associated with breast cancers” was provided.

Some additional comments:

A more nuanced treatment of zinc supplementation for cancer prevention is warranted. The results, if any are known, of the clinical trials cited in this section (lines 194-196) should be discussed. If breast cancers sequester and use more zinc than normal cells, is supplementing the diet a wise decision?

Author Response

Please see attached letter

Round 2

Reviewer 3 Report

I appreciate the author's attempts to address the stated concerns. However, the major concern that the section “Zinc associated with the CSC”  doesn’t present a focused hypothesis or a compelling rationale remains. In fact, adding so much information to the tangential points (author's response points 3-5) has made the rationale even more diffuse. In fact, this section now focuses on the role of zinc signaling in CCM, which is not the topic of this review. Therefore, revision is needed to focus and refine this section to, as was previously suggested:  clarify 1) the speculative hypothesis being proposed (presumably zinc regulation of the CmPn/CmP signaling network in breast cancer)and 2) the individual lines of evidence that support this hypothesis. In this case, the connection of zinc signaling to CCM lesion development is irrelevant.

Author Response

Please see attched letter
